# Targeted Delivery of Gene Silencing in Fungi Using Genetically Engineered Bacteria

**DOI:** 10.3390/jof7020125

**Published:** 2021-02-09

**Authors:** Jonatan Niño-Sánchez, Li-Hung Chen, Jorge Teodoro De Souza, Sandra Mosquera, Ioannis Stergiopoulos

**Affiliations:** 1Department of Plant Pathology, University of California Davis, Davis, CA 95616, USA; jnino@ucdavis.edu (J.N.-S.); lhchen@ucdavis.edu (L.-H.C.); jorge.souza@dfp.ufla.br (J.T.D.S.); smosquera@ucdavis.edu (S.M.); 2Department of Microbiology and Plant Pathology, University of California Riverside, Riverside, CA 92521, USA; 3Department of Plant Pathology, National Chung-Hsing University, Taichung 40227, Taiwan; 4Department of Plant Pathology, Federal University of Lavras (UFLA), Lavras, MG 37200-000, Brazil; 5Department of Ciencias Biológicas, Universidad EAFIT, Medellín 050022, Colombia

**Keywords:** RNA interference, HIGS, SIGS, dsRNA, cross-kingdom RNAi, *Escherichia coli* HT115(DE3), *Botrytis cinerea*, *Aspergillus flavus*, aflatoxins, bacterial autolysis

## Abstract

Exploiting RNA interference (RNAi) in disease control through non-transformative methods that overcome the hurdle of producing transgenic plants has attracted much attention over the last years. Here, we explored such a method and used non-pathogenic bacteria as a versatile system for delivering RNAi to fungi. Specifically, the RNaseIII-null mutant strain of *Escherichia coli* HT115(DE3) was transformed with two plasmid vectors that enabled the constitutive or IPTG-inducible production of double-stranded RNAs (dsRNAs) against genes involved in aflatoxins production in *Aspergillus flavus* (*AflC*) or virulence of *Botrytis cinerea* (*BcSAS1*). To facilitate the release of the dsRNAs, the bacterial cells were further genetically engineered to undergo a bacteriophage endolysin R-mediated autolysis, following a freeze-thaw cycle. Exposure under in vitro conditions of *A. flavus* or *B. cinerea* to living bacteria or their whole-cell autolysates induced silencing of *AflC* and *BcSAS1* in a bacteria concentration-dependent manner, and instigated a reduction in aflatoxins production and mycelial growth, respectively. In planta applications of the living bacteria or their crude whole-cell autolysates produced similar results, thus creating a basis for translational research. These results demonstrate that bacteria can produce biologically active dsRNA against target genes in fungi and that bacteria-mediated RNAi can be used to control fungal pathogens.

## 1. Introduction

RNA interference (RNAi) is a conserved mechanism of gene-silencing in nature with immense therapeutic potential. It is triggered by inverse transcripts that arecomplementary to the mRNA of a target gene, including double-stranded RNAs (dsRNAs) and small-interfering RNAs (siRNAs) that are produced by the DICER-mediated cleavage of dsRNAs into small pieces of 19–25 nucleotides [1]. Since its discovery, RNAi has been transformative to the study of gene function in numerous eukaryotic organisms and has created immense opportunities for its exploitation in medicine and disease control [2,3]. RNAi therapeutics is currently a rapidly evolving biotechnological field and despite the many challenges encountered along the way, 2018 marked a new era with the approval by the US Food and Drug Administration (FDA) of the first RNAi-based drug [2,4]. At present, several RNAi-based drugs that target diverse and rare human diseases are currently under development or undergoing phase I or phase II clinical trials. In addition, much effort is placed on the concomitant development of delivery strategies to target cells and tissues, which remains a key issue in the exploitation of RNAi therapeutics [2,5,6].

In crop protection, RNAi has mostly been utilized in the form of host-induced gene silencing (HIGS), an approach in which plants are genetically engineered to produce dsRNAs or siRNAs against genes in target pests and pathogens [7]. Several studies have highlighted the utility of HIGS in a variety of crops for the control of a wide-range of insect pests, nematodes, viruses, and fungal plant pathogens [7,8,9,10]. HIGS has also proven to be an effective method for reducing mycotoxin contamination in food and feed, including of the notorious aflatoxins [8,11,12,13,14,15]. Aflatoxins are secondary metabolites produced mainly by species of *Aspergillus* spp. and particularly by *A. flavus* and *A. parasiticus* [16,17]. At least 20 different aflatoxins have been described but aflatoxin B1 (AFB1), B2 (AFB2), G1 (AFG1) and G2 (AFG2) are the four most common types, with AFB1 and AFB2 mainly produced by *A. flavus*, and AFG1 and AFG2 by *A. parasiticus* [16,17]. They are highly toxic mycotoxins, well-known for their carcinogenicity and other serious illnesses related to liver damage and immunosuppression that they can cause in humans and livestock [18]. They enter the food chain mainly through contaminated agricultural commodities, with grain and nut crops such as maize, rice, and peanuts being major sources of contamination [17]. Estimates by the Food and Agriculture Organization (FAO) are that over 25% of the global food supply is contaminated by mycotoxins, including aflatoxins, whereas according to the World Health Organization (WHO), approximately 4.5 billion people are chronically exposed through their diet to these poisonous compounds, with often detrimental effects [16,17]. Thus, finding effective ways to mitigate contamination of food and feed by aflatoxins is crucial.

Despite the potential of HIGS for disease control, off-target effects, difficulties in plant transformation, and low public acceptance of transgene technology have restricted its application in the field [7,15,19,20]. Virus-induced gene silencing (VIGS), which exploits recombinant viruses as vectors to deliver RNAi molecules to plants, is an alternative to HIGS but to date it has mostly been utilized as a functional tool to study genes of interest in different crops, rather than for disease control [21,22,23]. The relatively recent discovery that externally applied dsRNAs and siRNAs can be taken-up by fungal cells from their environment and induce silencing in the transfected cells, has attracted much attention for its potential use as an alternative to HIGS for delivering RNAi to target pathogens [24]. Thus, spray-induced gene silencing (SIGS) has recently emerged as a novel, environmentally-friendly disease control method. At present, several attempts to generate RNAi-based biopesticides that utilize synthetic dsRNAs or siRNAs against genes in target pests and pathogens are underway [25,26]. Such RNA-based biopesticides can be customized to be species-specific or to target a broad range of pathogens, even across different taxonomic groups. Moreover, multiple genes can be simultaneously silenced, thus reducing the risk for resistance development [25,26,27]. However, a major limitation to the use of synthetic RNAi molecules as biopesticides is the cost of their large-scale production and formulation as well as the limited stability of the naked RNAi molecules under field conditions [26,27]. To overcome these limitations, a variety of synthetic materials have been developed to encapsulate and protect dsRNA from nuclease degradation. These include a double-layered hydroxide clay nanosheet that has been proved effective to protect dsRNA targeted against some viral diseases, or carbon nanotubes that successfully deliver siRNA and trigger posttranscriptional gene silencing in plant cells. Nevertheless, the use of these synthetic materials is still limited [3,28].

Here, we explored the possibility of using non-pathogenic bacteria as a versatile production and delivery system of RNAi to fungi. Bacteria-mediated delivery of RNAi has been successfully used to silence genes in *Caenorhabditis elegans* [29], insects [27], shrimps [30], and mammalian cells [31] but the utility of this approach for the control of fungi, to our knowledge, has not been exploited yet [26]. In our approach, a mutant of *Escherichia coli* lacking RNaseIII (an enzyme that degrades dsRNAs in bacterial cells) [29] was genetically engineered to express dsRNAs against *A. flavus* [32,33], and *Botrytis cinerea*, a notorious plant pathogen with a remarkably broad host range [34], and to further undergo a programmable autolysis [35], thus allowing the destruction of the bacterial cells and the release of the fungicidal dsRNAs. Our data show that the engineered bacteria accumulated high levels of dsRNAs, whereas subsequent treatment under in vitro or in planta conditions of *A. flavus* and *B. cinerea* with the living bacteria or the induced bacterial autolysates resulted in silencing of the complementary fungal genes and a reduction in production of aflatoxins or fungal growth, respectively.

## 2. Materials and Methods

### 2.1. Plasmid Construction

Plasmids pAGM4723-*AflC_250_* and pAGM4723-*eGFP_350_* (Appendix A) were constructed using modular cloning with the Golden Gate gene assembly [36,37]. Briefly, sense and antisense strands of *AflC* were PCR-amplified from the cDNA of this gene with primers pairs AflC-F/AflC-R and AflCr-F/AflCr-R, respectively (Appendix A). The *pKan^R^* promoter and *rrnB-T1* terminator were obtained from plasmid pPROBE-*gfp*[tagless] [38] by PCR amplification with primer pairs Ppntii-F/Ppntii-R and rrnB_T1-term-F/rrnB_T1-term-R, respectively (Appendix A). Sense and antisense strands of *AflC* were each cloned into a pICH41331 (Addgene MoClo Toolkit, Watertown, MA, USA) [36,37], in addition to *pKan^R^* and *rrnB-T1,* by the mixture of BpiI (Thermo Fisher Scientific, Waltham, MA, USA) and T4 DNA ligase (New England Biolabs, Ipswich, MA, USA). The assemblies were done in a way that *pKan^R^* and *rrnB-T1* flanked the *AflC_250_* fragments. Subsequently, the promoter-sense *AflC_250_*-terminator and promoter-antisense *AflC_250_*-terminator assemblies were cloned into pICH47732 and pICH47742 (Addgene MoClo Toolkit, Watertown, MA, USA) [36,37], respectively, using BsaI instead of BpiI in the mixture. The endolysin *R* gene containing the *pbla* promoter and *rrnB-T1* terminator was PCR-amplified from plasmid pAD-LyseR (Addgene Plasmid #99244) [35] with the primer pair EndoR_F/EndoR_R (Appendix A) and cloned into pICH47751 (Addgene MoClo Toolkit, Watertown, MA, USA) [36,37] by BsaI and T4 DNA ligase. Finally, the *AflC_250_* dsRNA synthesis cassette and endolysin *R* gene were assembled and cloned into pAGM4723 (Addgene MoClo Toolkit, Watertown, MA, USA) [36,37], using BpiI. For cloning pAGM4723-*eGFP_350_*, sense and antisense strands of *eGFP* were PCR-amplified from pICH41531 with primer pairs EGFP342-F/EGFP342-R and EGFP342-F-reversed/EGFP342-R-reversed, respectively (Appendix A). pAGM4723-*eGFP_350_* was constructed by using the procedure described above.

Plasmids pAGM4723-*BcSAS1_250_* and pAGM4723-*eGFP_250_* (Appendix A) were constructed by adding a T7 promoter (5′–TAATACGACTCACTATAG–3′) in the forward and reverse primers in opposite directions. *BcSAS1_250_* was PCR-amplified from the cDNA of *B. cinerea* with the primer pair BcSAS1_T7_F/BcSAS1_T7_R (Appendix A) and cloned into pICH47732 using BsaI and T4 DNA ligase in the reaction of the Golden Gate assembly. The *BcSAS1_250_* dsRNA synthesis cassette and endolysin *R* gene were assembled into pAGM4723 with BpiI and T4 DNA ligase. *EGFP_250_* was amplified from pAGM4723-*eGFP_350_* with primer pair G250_T7_F/G250_T7_R (Appendix A) and subsequently following the same procedure described above to clone it into pAGM4723.

### 2.2. Bacterial and Fungal Strains Used in This Study, Their Growth Conditions, and In Vitro Bioassays

The bacterial delivery system was based on the RNaseIII-deficient *E. coli* strain HT115(DE3). The genotype of HT115 is *E. coli* [F-, mcrA, mcrB, IN(rrnD-rrnE)1, rnc14::Tn10(DE3 lysogen: lacUV5 promoter-T7 polymerase] [29]. This strain originates from the W3110 strain, a commonly used wild-type of the non-pathogenic *E. coli* K-12 strain. The HT115(DE3) strain was selected for its ability to produce stable dsRNA and to express genes placed under the control of promoters recognized by the IPTG-inducible T7 RNA polymerase. The fungal strains analyzed were *A. flavus* NRRL3357 and *B. cinerea* B05.10. Fungal cultures were established from frozen mycelia stored in 25% glycerol (*v*/*v*) at −80 °C and maintained on potato dextrose agar (Difco Laboratories, Detroit, NI, USA) (PDA) plates.

*E. coli* HT115(DE3) was incubated in Luria-Bertani (LB) liquid medium [10 g Bacto^TM^ Tryptone (Difco Laboratories, Detroit, NI, USA), 10 g NaCl, 5 g Bacto^TM^ Yeast Extract (Difco Laboratories, Detroit, NI, USA), 1 L H_2_O, adjusted to pH 7.0] supplied with tetracycline (10 μg/mL) and kanamycin (50 μg/mL) at 37 °C at 300 rpm. The bacterial transformants HT115/*AflC*_Con_, HT115/*eGFP*_Con_, HT115/*BcSAS1*_Ind_ and HT115/*eGFP*_Ind_ were generated by the heat shock method in the presence of the plasmids pAGM4723-*AflC_250_,* pAGM4723-*eGFP**_350_**,* pAGM4723-*BcSAS1_250_,* and pAGM4723-*eGFP_250_*, respectively. Transformants were always grown in the presence of kanamycin (50 μg/mL) and when cultures reached an optical density at 600 nm (OD_600_) of approximately 0.8, then isopropyl-β-D-1-thiogalactopyranoside (IPTG) (Sigma-Aldrich, St. Louis, MO, USA) was added at 1.0 mM final concentration to the cultures of HT115/*BcSAS1*_Ind_ and HT115/*eGFP*_Ind_ to induce dsRNA production when needed. Bacterial concentration was measured with the Spectronic Genesys 6 spectrophotometer (Thermo Electron Scientific Instruments LLC, Madison, WI, USA) as optical density at 600 nm (OD_600_). Bacterial cultures were incubated until they reached the stationary phase, which was experimentally determined to be at an OD_600_ of 1.2–1.3 (Appendix A). The volumes corresponding to a final OD_600_ of 1, 0.5, 0.1, and 0.01 in 10 mL cultures were concentrated by centrifugation (3000× *g*, 5 min) and re-suspended in molecular grade water (G-Biosciences, St. Louis, MO, USA) to a final volume of 0.5 mL. To induce autolysis, the bacterial cells were subjected to a freeze-thaw cycle consisting of 15–20 min at −80 °C, followed by thawing for 60–90 min at 37 °C and subsequently treating the cell suspensions with the antibiotics ampicillin (100 μg/mL final concentration) and streptomycin (50 μg/mL) for 1 h, to assure that no living cells were present. Growth curves for the HT115/*AflC*_Con_, HT115/*eGFP*_Con_, HT115/*BcSAS1*_Ind_, and HT115/*eGFP*_Ind_ strains as well as the HT115(DE3) strain (Appendix A), were calculated by measuring the OD_600_ with the Genesys 6 spectrophotometer (Thermospectronic) every hour up to 9 h of growth, and then at 13 h and 24 h. All strains were grown in LB medium supplemented with tetracycline (10 μg/mL) and kanamycin (50 μg/mL). Kanamycin was not added to the HT115(DE3) culture, whereas when reaching OD_600_ ≈ 0.8, strains HT115/*BcSAS1*_Ind_ and HT115/*eGFP*_Ind_ were induced with 1.0 mM IPTG. In each of the three experiments performed, the cultures were grown at 37 °C in 20 mL volumes with 250 rpm of shaking, with three replicates per strain.

To obtain fungal spores, *A. flavus* NRRL3357 was grown on PDA medium with continuous light at 37 °C for 3–5 days (non-aflatoxin production conditions), while *B. cinerea* was grown on PDA medium at 25 °C for 10–14 days. Spores were collected by scrapping mycelia with 2 mL of milli-Q water and filtrating them through two layers of cheesecloth. Spore concentration was measured with a hemocytometer. The spore suspension was subsequently added to 5 mL of YES liquid medium [2% Bacto^TM^ Yeast Extract (Difco Laboratories, Detroit, NI, USA), 6% sucrose, pH 5.8] to a final concentration of 10^4^ spores/mL. The spore suspension was then incubated at 37 °C (*A. flavus*) or 25 °C (*B. cinerea*), 250 rpm with continuous light. Once fungal spores had germinated (checked by microscopy at 6–8 h), the bacterial treatments were added to the fungal culture flasks. These bacterial treatments were either the living bacterial cells or their whole-cell autolysates produced in 10 mL and concentrated in 0.5 mL, as described above. The co-incubation conditions for both fungi were 25 °C, 250 rpm, in darkness. Fungal samples were harvested 12 h, 24 h, and 36 h after the bacterial treatment applications.

*B. cinerea* dry weight was measured after filtering the fungal culture with a vacuum filtration system using a 0.45 μm pore size nitrocellulose membrane (Thermo Fisher Scientific, Waltham, MA, USA). Once samples were completely dry, they were fully removed from the membrane with the help of a spatula and measured using a precision mg scale (Mettler Toledo, ML203E/03). Culture filtering by vacuum filtration, performed as described above, was also used to harvest the supernatant of *A. flavus* cultures, of which 1 mL was used to measure total content of aflatoxins by Enzyme-Linked Immunosorbent Assay (ELISA). Dried fungal mycelium was frozen in liquid nitrogen and stored at −80 °C before total RNA extraction with the TRizol^TM^ reagent (Invitrogen, Carlsbad, CA, USA).

### 2.3. Total RNA Extractions and RT-qPCR

Total RNA was extracted from *A. flavus*, *B. cinerea,* the inoculated plant tissues, and the bacterial strains using the TRIzol^TM^ reagent (Invitrogen, Carlsbad, CA, USA), according to the manufacturer’s instructions. Briefly, 1 mL of TRIzol^TM^ reagent (Invitrogen, Carlsbad, CA, USA) was added to 50–100 mg of ground plant tissue, fungal, or bacterial cells, and incubated for 5 min at room temperature. Then, 0.2 mL of chloroform was added to the samples and incubated for 3 min at room temperature, followed by centrifugation for 15 min at 12,000× *g* at 4 °C. The nucleic acids in the supernatant were precipitated by adding 500 μL of isopropyl alcohol (1:1) and centrifuging at 12,000× *g* at 4 °C for 10 min. Pellets were washed with 75% ethanol/water (*v*/*v*), re-suspended with DEPC-treated water (Invitrogen, Carlsbad, CA, USA), and treated with DNaseI according to the manufacturer’s instructions (Thermo Fisher Scientific, Waltham, MA, USA). Following total RNA extraction, samples were tested for integrity in 1.2% agarose gels and quantified using a Nanodrop Spectrophotometer (Nanodrop 2000, Thermo Fisher Scientific, Waltham, MA, USA). The absence of DNA contamination was examined in the samples by the lack of PCR amplification of the glyceraldehyde 3-phosphate dehydrogenase gene (*GADPH*) gene in *A. flavus* and of the *BcActin* gene in *B. cinerea*. cDNA was synthesized from purified total RNA using iScript^TM^ cDNA Synthesis Kit (Bio-Rad Laboratories, Hercules, CA, USA), following the manufacturer’s instructions. The SYBR Green kit SsoFast^TM^ EvaGreen^®^ (Bio-Rad Laboratories, Hercules, CA, USA) was used to carry out RT-qPCR reactions in a CFX96 Touch thermal cycler (Bio-Rad laboratories, Hercules, CA, USA) equipped with the CFX Maestro^TM^ software. The thermal profile was as recommended by the manufacturer (40 cycles of 98 °C for 3 s and 60 °C for 5 s), testing single amplification by a specific peak in the dissociation curve (0.5 °C increments every 10 s within a range of 65–95 °C). The *GADPH* gene, amplified with the primer pair qGADPH-F/qGADPH-R (Appendix A), and the *BcActin* gene, amplified with the primer pair qBcActin-F/qBcActin-R (Appendix A), were used as endogenous reference genes for *A. flavus* and *B. cinerea,* respectively. The primer pair qAflC-F/qAflC-R (Appendix A) was used for *AflC* amplification, while primer pair qBcSAS1-F/qBcSAS1-R (Appendix A) was used for *BcSAS1*. The *AflC* and *BcSAS1* sequences of the amplicons generated by RT-qPCR do not match with the dsRNA sequence used in the treatments, to avoid artifacts. The relative expression level of each gene was calculated by the 2^−ΔΔCt^ method [39].

### 2.4. dsRNA Isolation from E. coli HT115(DE3)

Total RNA was extracted from *E. coli* HT115(DE3) cultures using the TRIzol^TM^ reagent (Invitrogen, Carlsbad, CA, USA) according to the manufacturer’s instructions and as described above. Then, dsRNAs were isolated from total RNA extraction, using one of the following two methods. In the first method, 20 μg of total RNA was treated at 37 °C for 45 min with 3 μL of DNaseI (Thermo Fisher Scientific, Waltham, MA, USA) and 1 μL of RNase A (Thermo Fisher Scientific, Waltham, MA, USA) in a 50 μL reaction volume containing 0.35 mM NaCl and 5 μL of 10 × DNase I buffer. dsRNAs were further purified with the RNA Clean & Concentrator Kit (ZYMO Research, Irvine, CA, USA) or the mirVana^TM^ miRNA isolation kit (Thermo Fisher Scientific, Waltham, MA, USA), following the specifications for <200 nucleotides RNA removal. In the second method, the MEGAscript^TM^ T7 Transcription Kit (Thermo Fisher Scientific, Waltham, MA, USA) was used from the step of nuclease digestion to remove DNA and ssRNAs from the filter cartridge provided with the kit, followed by purification of dsRNAs from the filter cartridge, which is able to remove free nucleotides and <200 nucleotides nucleic acid degradation products. Finally, dsRNAs were visualized by electrophoresis and their concentration was determined with a Nanodrop Spectrophotometer (Thermo Fisher Scientific, Waltham, MA, USA).

To examine dsRNA stability over time in the bacterial autolysates and in living bacterial cultures, strain HT115/*BcSAS1_Ind_* was grown to the stationary phase (OD_600_ ≈ 1.2–1.3) in LB medium supplemented with tetracycline (10 μg/mL) and kanamycin (50 μg/mL). When the culture had reached an OD_600_ ≈ 0.8, IPTG was added to a final concentration of 1.0 mM. The culture was subsequently diluted to an OD_600_ of 1.0 in 10 mL volumes. After that, the 10 mL cultures were pelleted by centrifugation at 3000× *g* for 5 min and re-suspended in 0.5 mL of molecular grade water (G-Biosciences, St. Louis, MO, USA). The bacterial autolysates were produced as described above. Living bacterial cells were produced without the freeze-thaw cycle and the antibiotic treatment. Subsequently, the autolysates and the living bacterial cells were incubated at room temperature (≈ 25 °C) for 0 h, 1 h, 6 h, 24 h, 36 h, and 72 h, after which the debris was removed by centrifugation (21,000× *g*, 30 min, at 4 °C) and total nucleic acids were extracted from the supernatants with the TRIzol^TM^ method as described above. The nucleic acids were then treated with RNaseA and DNase I, and cleaned with the RNA Clean & Concentrator™ Kit, as described above. Finally, aliquots of 0.5 μg of RNA were loaded onto 2% agarose gels for electrophoresis.

### 2.5. Bioassays

Fresh maize seeds were surface sterilized as described previously [40]. Briefly, the entire seeds were treated with a 75% ethanol/water (*v*/*v*) solution for 30 s at room temperature, then rinsed with a milli-Q autoclaved water. Next, they were treated with a 2% hypochlorite solution for 5 min at room temperature and rinsed at least five times with milli-Q autoclaved water. Aliquots of 2 μL of a 50 × 10^6^ spore/mL suspension (milli-Q autoclaved water) of *A. flavus* were placed on a maize embryo and were incubated at 37 °C with light for 2 days (non-aflatoxin production conditions). After this time period, 10 μL of living or lysed bacteria cells obtained from the OD_600_ 1.0 culture were placed at the same point of fungal inoculation and incubated for one more day at aflatoxin-production conditions (darkness at 25 °C). Humidity was maintained to over 80% by placing the maize seeds into boxes layered with sterile Whatman filter paper 1/5, which was hydrated with 1 mL of sterile water every 24 h. Subsequently, the maize embryos with the fungal mycelium were harvested for further analysis with RT-qPCR (40 cycles of 98 °C for 3 s and 60 °C for 5 s) and ELISA.

*Nicotiana benthamiana* leaves were inoculated with *B. cinerea,* as described previously [41]. Briefly, *B. cinerea* spores were collected from PDA plates incubated for 10–14 days, as described above. Spores were purified and re-suspended at a concentration of 10^4^ spores/mL in grape juice (1/2 strength). A drop of 4 μL of this spore suspension was inoculated onto both sides of detached 4–6 week-old *N. benthamiana* leaves, which were placed onto 1.5% (*m*/*v*) phytoagar (Thomas Scientific, Swedesboror, NJ, USA) medium trays. After 6 h, 10 μL of living or lysed bacteria cells obtained from the OD_600_ 1.0 culture were placed at the same point of fungal inoculation, on the left side of the leaf blade for HT115/*eGFP_Ind_* and the right side of the leaf blade for HT115/*BcSAS1_Ind_*. Suspensions of living bacteria cells were applied with IPTG at a final concentration of 0.1 mM. Inoculated *N. benthamiana* leaves were incubated for 3 days. The relative lesion size between treatments in each leaf was determined by measuring the necrotic area using the ImageJ software. Then, they were collected for RT-qPCR analysis (40 cycles of 98 °C for 3 s and 60 °C for 5 s).

### 2.6. Analysis of Total Aflatoxins by ELISA

The total concentration of aflatoxins on *E. coli*-*A. flavus* co-cultures and maize seeds infected with *A. flavus* was analyzed by ELISA. The AgraQuant^®^ Total Aflatoxin Test Kit (4–40 ppb) (Romer labs^®^, Tulln, Austria) was used for the ELISA analysis according to the manufacturer’s instructions. Briefly, aflatoxins were extracted with 70% (*v*/*v*) methanol from 1 mL of fungal culture supernatants or grinded inoculated maize seeds. Because the concentration of aflatoxins in the samples was unknown and it could fall outside of the range of the solutions of known aflatoxin concentration provided by the manufacturer (i.e., the standards), serial dilutions of the experimental samples were prepared (1:0, 1:10, 1:100, and 1:1000) in 70% (*v*/*v*) methanol, so that one of the serial dilutions would fall within the aflatoxin concentration range covered by the standards provided with the manufacture’s kit (0, 4, 10, 20, 30 and 40 ppb). The OD of each sample dilution at 450 nm (OD_450_) was measured at the end of the ELISA reactions and the sample dilution whose OD_450_ measurement was in the range of the manufacturer’s standards was selected. Subsequently, aliquots of 100 μL of the selected diluted samples and the standards in duplicate were added to 200 μL of the conjugate. After mixing, 100 μL were transferred to antibody-coated wells and incubated for 15 min at room temperature. The unbound conjugate was removed with deionized water during the washing step (5×). Once wells were dry, 100 μL of substrate were added into each well, incubated for 5 min, and then 100 μL of the stop solution were added. Microtiter plates were measured in an ELISA reader (Tristar LB941, Berthold Technologies, Bad Wildbad, Germany) at 450 nm. The concentration of aflatoxins in the samples was calculated using the formula of the calibration curve from the standards and multiplied by the specific dilution factor of the sample.

### 2.7. Statistical Analyses

Results represent the mean and the standard deviations of at least three biological replicates, which are indicated by error bars. Significant differences were determined using an unpaired *t*-test to compare the means between two groups, and a one-way ANOVA followed by Tukey’s Honestly Significant Difference (HSD) test to compare the means between three or more groups. The one-sample *t*-test was used to test differences between the relative aflatoxins production or fungal lesion size of each biological treatment with respect to the control treatment (designated as 1). ^o^
*p* < 0.10; * *p* < 0.05; ** *p* < 0.01, *** *p* < 0.001. Statistical analyses were performed using Statistix 10.0 software package (Analytical Software, Tallahassee, FL, USA).

## 3. Results

### 3.1. Bacteria Can Be Genetically Engineered to Produce dsRNAs against Target Genes in Fungi

For our studies, we have utilized the well-characterized *E. coli* HT115(DE3) strain, which lacks the *rnc* allele that encodes for the dsRNA-specific endonuclease RNaseIII, and is thus capable of synthesizing and preserving dsRNAs. The strain is also lysogenic for λDE3, thus enabling the transcription of genes placed under the control of the bacteriophage T7 promoter [29]. HT115(DE3) was genetically engineered to express dsRNAs against *AflC* in *A. flavus* [42] and *BcSAS1* in *B. cinerea* [43]. These two pathogens were selected due to their importance as a producer of aflatoxins and a plant pathogen with a broad host range, respectively [32,34]. In addition, silencing following uptake of dsRNAs by their mycelia has been demonstrated for both fungi [12,13,15,24,40]. Likewise, *AflC* and *BcSAS1* were selected as gene targets for RNAi because of their critical role in production of aflatoxins in *A. flavus* and virulence in *B. cinerea,* respectively. Specifically, *AflC* encodes a Type I polyketide synthase (pksA) needed for the synthesis of the backbone structure of aflatoxins in *A. flavus*, whereas *BcSAS1* encodes for a member of the Rab family of small GTPases that is required for mycelial development and virulence in *B. cinerea* [42,43].

To further explore the versatility of the system, dsRNAs against *AflC* and *BcSAS1* were produced in HT115(DE3) using two plasmid vectors that allowed the production of dsRNAs via different transcriptional configurations, i.e., (i) through the constitutive in trans (i.e., from separate plasmid loci) transcription of the sense and antisense strands of the selected target DNA template (Figure 1A) and (ii) through the IPTG-inducible in cis (i.e., from the same plasmid locus) bi-directional transcription of the DNA template (Figure 1B). Specifically, in the first configuration that was used to produce dsRNAs against *AflC*, a 250 bp cDNA fragment was cloned in opposite directions between the native promoter of the kanamycin resistance gene (*pKan^R^*) and the transcriptional terminator T1 of the *E. coli rrnB* gene (*rrnB-T1*). A plasmid construct similar to this one but in which the *AflC* fragment was replaced by a 350 bp cDNA fragment of the enhanced green fluorescent protein (eGFP) was also prepared and used in control applications (Figure 1A and Appendix A). In the second configuration that was used to produce dsRNAs against *BcSAS1*, two T7 promoters (*pT7*) were placed in an inverted orientation at the 5′ and 3′-end of a 250 bp cDNA fragment of the gene (Figure 1B). As for *AflC*, a plasmid construct in which the *BcSAS1* fragment was replaced by a 250 bp *eGFP* cDNA fragment was also prepared and used as control (Figure 1B and Appendix A). Following transformation into HT115(DE3) of the plasmid constructs and transcription, the first configuration produces in trans complementary sense and antisense ssRNA molecules from the DNA templates, which subsequently hybridize to form dsRNAs (Figure 1C). By contrast, in the second configuration sense/antisense transcript pairs are spontaneously formed in cis, thus creating dsRNAs (Figure 1C). Moreover, while in the first configuration the transcription of the target sequences is controlled by the constitutive *pKan**^R^* promoter, in the second configuration it is dependent on the induction of the *T7* gene 1 in HT115(DE3) by IPTG and subsequent activation of the T7 promoter by the T7 RNA polymerase.

As both trans-kingdom transfer and environmental uptake of RNAi molecules by mycelial cells have been demonstrated to induce RNAi in fungi, in order to combine the benefits of the two approaches, we further genetically engineered the HT115(DE3) strain to allow the release of the dsRNA cargo through a programmable cellular autolysis step. For this purpose, a phage lambda endolysin gene (*λR*), placed under the control of the weakly constitutive *pbla* promoter of the β-lactamase gene and the *rrnB-T1* terminator was cloned in the two plasmid vectors used for the production of the dsRNAs (Figure 1A,B and Appendix A). When the inner bacterial cell membrane is intact, the endolysin protein accumulates in the cell without significantly impacting cell growth. However, a simple freeze-thaw cycle opens the inner membrane, thus enabling the endolysin to disrupt the bacterial cell wall and trigger autolysis [35]. Notably, although cell lysis results in the release of several lytic enzymes, the autolysis of the bacterial cell did not trigger a parallel degradation of the dsRNAs, which remained stable in the cell lysates up to 24 h, and partially stable up to 72 h of incubation at 25 °C. If the cells are not lysed, then dsRNAs remain stable in the living cells for at least for 72 h (Figure 1D). This indicates that both living bacteria and their lysates can be used to deliver dsRNAs against fungi.

### 3.2. Bacterially-Produced dsRNAs Induce RNAi in Fungi in a Bacteria Concentration-Dependent Manner

We initially sought to determine whether the HT115(DE3) bacterial strains that were genetically engineered to produce dsRNAs against *AflC* [HT115(DE3)(pAGM4723-*AflC_250_*)], hereafter abbreviated as HT115/*AflC_Con_**,* or *BcSAS1* [HT115(DE3)(pAGM4723-*BcSAS1*_250_)], hereafter abbreviated as HT115/*BcSAS1_Ind_**,* were able to induce silencing of the targeted transcripts in *A. flavus* and *B. cinerea,* respectively. Two approaches for delivering the bacterially-produced dsRNAs against *A. flavus* and *B. cinerea* were utilized, i.e., trans-kingdom mediated transfer, facilitated by co-incubating living bacteria with fungal germlings (hereafter referred to as Treatment A), and environmental uptake from bacterial whole-cell autolysates (hereafter referred to as Treatment B).

To examine whether the physical interaction between fungi and bacteria can facilitate the trans-kingdom transfer of the bacterially-produced dsRNAs to fungi (Treatment A), 10 mL bacterial cultures with a final OD_600_ of 1, 0.5, 0.1, and 0.01 that were prepared from serially-diluted cultures at stationary phase (OD_600_ ≈ 1.2–1.3) (Appendix A), were concentrated to a volume of 0.5 mL and mixed with 5 mL of fungal germlings (10^4^ germlings/mL) in liquid cultures. Assessment of RNAi-induced gene silencing at 12 h of co-incubation between *A. flavus* and the HT115/*AflC_Con_* strain, or between *B. cinerea* and the HT115/*BcSAS1_Ind_* strain (hereafter referred to as test applications), showed a significant reduction in *AflC* and *BcSAS1* transcript levels compared with the expression levels of these two genes when the two fungi were co-incubated with the HT115/*eGFP_Con_* and HT115/*eGFP_Ind_* strains, respectively (hereafter referred to as control applications) (Figure 2A,B, and Appendix A). Overall, when germlings of *A. flavus* or *B. cinerea* were co-incubated with the bacterial concentrates obtained from the OD_600_ 1.0, 0.5, and 0.1 cultures, a 21.7-fold, a 5.5-fold, and a 3.2-fold reduction in *AflC* expression, and a 23.3-fold, a 15.9-fold, and a 2.7-fold reduction in *BcSAS1* expression were observed in the test applications relative to the control ones, respectively (Figure 2A,B and Appendix A). These reductions in *AflC* and *BcSAS1* transcript levels between the test and control applications were significant at α = 0.05 (Appendix A) and positively correlated with the OD_600_ of the bacterial cultures (*AflC*: *r =* 0.74, *p =* 0.0061; *BcSAS1*: *r =* 0.73, *p =* 0.007). Moreover, the reduction in target gene expression was presumably caused by the bacterially-produced dsRNAs alone, as the expression levels of *AflC* and *BcSAS1* did not statistically differ among the control applications, or between the control applications and the axenic culture of *A. flavus* and *B. cinerea* (Appendix A). Taken together, the above results indicate that the two HT115(DE3) strains that were genetically engineered to produce dsRNAs against *AflC* or *BcSAS1* are capable of inducing RNAi to *A. flavus* and *B. cinerea* in in vitro co-cultures of the two organisms, in a bacteria concentration-dependent manner.

As for Treatment A, a bacteria concentration-dependent response in the silencing levels of *AflC* and *BcSAS1* (*AflC*: *r =* 0.80, *p =* 0.0017; *BcSAS1*: *r =* 0.77, *p =* 0.0037) was also observed when germlings of *A. flavus* and *B. cinerea* were incubated for 12 h under the same experimental conditions with the whole-cell autolysates of HT115/*AflC_Con_* and HT115/*BcSAS1_Ind_*, respectively (Treatment B) (Figure 2A,B and Appendix A). Specifically, a significant (*p* < 0.05) reduction in the expression levels of *AflC* in the test relative to the control applications was obtained when *A. flavus* was incubated with lysates of HT115/*AflC_Con_* collected from the OD_600_ 1.0 and 0.5 cultures but not with the lysates collected from the OD_600_ 0.1 and 0.01 cultures (Figure 2A and Appendix A). On a quantitative basis, subtle changes in *AflC* expression between test and control applications were observed in Treatment B compared with Treatment A for all bacterial densities tested (Appendix A), suggesting that environmental uptake of dsRNAs from bacterial whole-cell autolysates was less efficient in eliciting RNAi in *A. flavus* compared with applications of living bacteria. When considering *BcSAS1*, then a significant (*p* < 0.05) reduction in its expression level was observed when germlings of *B. cinerea* were co-incubated with lysates of HT115/*BcSAS1_Ind_* collected from all four serially diluted cultures (i.e., cultures with an OD_600_ of 1, 0.5, 0.1, 0.01) (Figure 2B and Appendix A). Moreover, fold-changes in *BcSAS1* transcript levels between test and control applications were lower in Treatment B compared with Treatment A for the OD_600_ 1.0 and 0.5 cultures, but higher for the OD_600_ 0.1 and 0.01 cultures (Figure 2B and Appendix A). This suggests that higher densities of cell lysates could be interfering with the absorption of dsRNAs by the hyphae, thus making it appear that living bacteria are more potent in eliciting silencing. Finally, the inclusion of just the bacterial lysates in the liquid cultures had no effect on the expression of *AflC* or *BcSAS1*, as their transcript levels were essentially the same among control applications (Appendix A).

In order to appreciate the silencing levels achieved against *AflC* and *BcSAS1* in Treatments A and B, an additional application of purified dsRNAs was evaluated, consisting of 15 μg (12.5 nM) of dsRNAs against *AflC* or *BcSAS1* applied to 5 mL of in vitro cultures (10^4^ germlings/mL) of *A. flavus* or *B. cinerea*, respectively. Compared with the expression levels of these two genes in control axenic cultures of *A. flavus* or *B. cinerea,* the application of the purified dsRNAs led to a 2.9-fold reduction in *AflC* transcript levels (Figure 2A) and an 8.9-fold reduction in *BcSAS1* transcript levels (Figure 2B). These changes in gene expression are intermediate between those obtained in Treatments A and B at a bacterial OD_600_ of 0.1 and 0.5. Thus, collectively our results indicate that the bacterial-based platform to produce dsRNAs can be effectively utilized to silence fungal genes, achieving results that are on par with those obtained with applications of naked dsRNAs.

### 3.3. Increased Contact Times between Fungi and dsRNA-Producing Bacteria Have a Positive Effect on Reducing Production of Aflatoxins in A. flavus and Mycelial Growth in B. cinerea

We next assessed whether an increase in contact time between HT115/*AflC_Con_* and *A. flavus* or between HT115/*BcSAS1_Ind_* and *B. cinerea*, would have an added positive effect on the dsRNA-mediated induced silencing of *AflC* and *BcSAS1*, respectively. For this purpose, concentrates of living bacteria or of their whole-cell autolysates from the OD_600_ 1.0 cultures were added to 5 mL cultures of *A. flavus* or *B. cinerea*, and silencing levels of *AflC* and *BcSAS1* were assessed after 12 h, 24 h, and 36 h of co-incubation (Appendix A). All other conditions in these experiments were kept the same as in the in vitro experiments described above, including control applications.

Examination of the *AflC* transcript levels in just the control applications showed that the expression of this gene decreased over time in both Treatment A and Treatment B. The changes in *AflC* transcript levels from 12 h to 24 h and 36 h were significant at α = 0.05 (12 h to 24 h, Treatment A and B) or α = 0.10 (12 h to 36 h, Treatment A). However, there were not significant differences in *AflC* expression levels between 24 h and 36 h for neither Treatment A nor Treatment B (Appendix A). When assessing *AflC* silencing levels in the test relative to the control applications, then in Treatment A, a large 21.7-fold decrease in the *AflC* expression levels was observed at 12 h, but only a 5.2-fold and a 4.6-fold decrease were observed at 24 h and 36 h, respectively. The decreases in *AflC* expression at 12 h and 24 h were significant at α = 0.05, whereas the decrease at 36 h was significant only at α = 0.1. In a similar way, in Treatment B, a 4.1-fold, a 1.7-fold, and a 2.6-fold decrease in *AflC* expression were seen at 12 h, 24 h, and 36 h, respectively in the test relative to the control applications, of which only the decrease at 12 h was significant at α = 0.05 (Figure 3A and Appendix A). Collectively, the data suggest that although the HT115/*AflC_Con_* living bacteria or of their lysates strongly repressed *AflC* transcript levels during the first 12 h, the silencing effect was nearly alleviated by 24 h, despite the continuous presence of the bacteria or their lysates in the co-cultures. However, it is also possible that the absence of detectable silencing of *AflC* at 24 h and 36 h post bacterial applications is caused by the low expression levels of this gene at these two time points that diminishes the effects of the dsRNA treatment.

Since *AflC* encodes for a key enzyme in the biosynthetic pathway of aflatoxins [42], we next examined whether the observed HT115/*AflC_Con_*-mediated silencing of *AflC* would influence their production. Quantification of aflatoxins content in co-cultures of Treatment A and B after 12 h, 24 h, and 36 h showed that, unlike *AflC* transcript levels, the total amount of these mycotoxins increased over time, most likely because of the parallel increase and accumulation in fungal biomass (Figure 3B and Appendix A). However, when comparing aflatoxins content between test and control applications of either Treatment A or B, then a decrease was observed in the test applications that was more pronounced at 36 h, followed by 24 h and 12 h. Specifically, in co-cultures of Treatment A, a 64.6% and a 30.5% reduction in aflatoxins content was observed in the test relative to the control applications at 36 h and 24 h of co-incubation time, respectively, whereas at 12 h, a 13.3% increase was observed. The decrease at 36 h was significant at α = 0.05, whereas at 24 h it was significant at α = 0.1 and at 12 h it was not significant (Figure 3B and Appendix A). In a similar way, in co-cultures of Treatment B, a 42.3%, 35.4%, and 0% reduction in aflatoxins content was observed in the test relative to the control application at 36 h, 24 h, and 12 h of co-incubation time, respectively. The differences were significant at α = 0.05 at 36 h and 24 h but not significant at 12 h (Figure 3B and Appendix A). Overall, our results show that bacterially-produced dsRNAs against *AflC* can significantly impair production of aflatoxins over time, whereas the inverse correlation seen between *AflC* mRNA abundance and concentration of aflatoxins, suggests that there is likely a delay between *AflC* expression, and biosynthesis and secretion of aflatoxins.

When examining how increased contact times between *B. cinerea* and living HT115/*BcSAS1_Ind_* or the lysates thereof affected *BcSAS1* transcript levels and subsequent growth of the fungus, then a positive correlation was observed as well. Overall, *BcSAS1* expression levels increased from 12 h to 24 h and 36 h in the control applications of both Treatment A and Treatment B. However, the differences in *BcSAS1* expression between 12 h, 24 h, and 36 h were not significantly different from each other (Appendix A), indicating that the expression of this gene was essentially stable in the controls throughout the course of the experiment. When compared with the control applications, exposure of *B. cinerea* to living HT115/*BcSAS1_Ind_* bacterial cells or their whole-cell autolysates resulted in a significant at α = 0.01 reduction in *BcSAS1* transcript levels at 12 h and 36 h of co-incubation time. However, at 24 h, the reduction was significant only at α = 0.10 (Figure 3C and Appendix A). Specifically, in co-cultures of Treatment A, a 23.3-fold and 14.5-fold decrease in *BcSAS1* transcripts levels were observed in the test applications relative to the control ones at 12 h and 36 h, respectively whereas only a 4.3-fold reduction was seen at 24 h. In a similar way, in co-cultures of Treatment B, a 20.0-fold and a 25.6-fold reduction were observed at 12 h and 36 h, respectively, whereas a 2.9-fold reduction was seen at 24 h (Figure 3C and Appendix A). These oscillations in *BcSAS1* silencing levels from 12 h to 24 h and 36 h could represent an attempt by the fungal cells to partially recover from the effects of RNAi, perhaps through the activation of a regulatory feedback mechanism that increased *BcSAS1* expression at around 24 h in an effort to compensate for the reduction in its mRNA levels. Irrespective of whether such a negative-feedback loop is operational for *BcSAS1*, overall, the results indicate that increased contact times between the fungus and the dsRNA-producing bacteria can have a positive impact on silencing of target genes, although this activity seems to be dependent on the gene target and possibly other parameters as well.

Since *BcSAS1* is involved in mycelial development in *B. cinerea* [43], we also examined whether the observed HT115/*BcSAS1_Ind_*-mediated silencing of *BcSAS1* at 12 h, 24 h, and 36 h of co-incubation time between the fungus and the living bacterial cells or their autolysates would impact fungal growth. Indeed, a 16.0%, 27.4%, and 24.1% reduction in fungal dry weight was observed in the test relative to the control co-cultures of Treatment A at 12 h, 24 h, and 36 h of co-incubation time between the fungus and the bacteria, respectively (Figure 3D and Appendix A). Although small, these changes were significant at α = 0.05 for 12 h and 24 h but not at 36 h (Appendix A). In a similar way, a 39.2%, 61.3%, and 24.1% reduction in fungal dry weight was seen at 12 h, 24 h, and 36 h, respectively in the test co-cultures of Treatment B relative to the control ones (Figure 3D and Appendix A). It is worth noticing that the oscillations observed in the reduction of fungal biomass at 12 h, 24 h, and 36 h inversely mirrored the fluctuations observed in the *BcSAS1* silencing levels at these time points, suggesting that there could be a delay between inhibition of mRNA and phenotypic expression of growth impedance. This was more pronounced in Treatment B, as fungal growth in Treatment A was partially inhibited by growth of the bacteria, thus alleviating some of the observed differences. In this respect, fungal growth as determined by mycelial dry weight in the control co-cultures of Treatment A was reduced by 36.1% at 12 h (*p =* 0.068), 41.7% at 24 h (*p =* 0.003), and 16.9% at 36 h (*p =* 0.213) compared with Treatment B (Appendix A). Despite these differences between Treatment A and Treatment B, overall the results demonstrate that applications of bacteria producing dsRNAs against *BcSAS1* can effectively reduce mycelial growth of *B. cinerea* under in vitro conditions.

### 3.4. In Vivo Applications of dsRNA-Producing Bacteria Result in a Reduction of Aflatoxins Production by A. flavus and Disease Symptoms Caused by B. cinerea

The in vitro experiments demonstrated that HT115/*AflC_Con_* and HT115/*BcSAS1_Ind_* could be used as vectors for delivering RNAi against the target genes in *A. flavus* and *B. cinerea*, and thus impede production of aflatoxins and mycelial growth, respectively. Therefore, we next examined whether such an effect could also be achieved under in vivo conditions during fungal infections of host plants. For this purpose, *A. flavus* and *B. cinerea* were spot inoculated on maize seeds and leaves of *N. benthamiana*, respectively and after a period of incubation of 48 h for *A. flavus* and 6 h for *B. cinerea,* 10 μL of living bacterial concentrates or their whole-cell autolysates from the OD_600_ 1.0 cultures were applied to the same fungal inoculation spots.

Examination of *AflC* transcript levels in *A. flavus* inoculated on maize seeds (Figure 4A) and exposed for 24 h at living HT115/*AflC_Con_* bacteria or their autolysates, showed a significant (α = 0.05) 4.8-fold and a 6.5-fold reduction, respectively in its expression levels in the test relative to the control applications (Figure 4B and Appendix A). Notably, *AflC* expression levels did not statistically differ between control applications of Treatment A and Treatment B (*p =* 0.441), indicating that silencing of *AflC* was caused by the dsRNAs produced by the HT115/*AflC_Con_* bacteria. Moreover, the decrease in *AflC* transcript levels was further accompanied by a reduction in production of aflatoxins, as determined by parallel measurements of total aflatoxins content in the maize seeds (Figure 4C and Appendix A). Specifically, for Treatment A, a 47.4% reduction was observed in the total amount of aflatoxins extracted from seeds treated with the HT115/*AflC_Con_* strain compared with seeds treated with the HT115/*eGFP_Con_* strain. In a similar way, for Treatment B, a 51.8% reduction in aflatoxins content was observed in seeds included in the test application relative to seeds included in the control one (Figure 4C and Appendix A). The reduction in the total aflatoxins content between test and control applications clearly demonstrates that the HT115/*AflC_Con_* strain can effectively obstruct the production of these mycotoxins by *A. flavus,* also under in vivo conditions.

As for *AflC*, application of the living HT115/*BcSAS1_Ind_* cells or their whole-cell autolysates on *N. benthamiana* leaves inoculated with *B. cinerea* (Figure 5A) resulted in silencing of *BcSAS1*. Indeed, a 2.6-fold and 1.7-fold reduction in *BcSAS1* expression was seen in the test applications relative to the control ones for Treatment A and Treatment B, respectively at 72 h post addition of the living bacteria or their whole-cell autolysates on *N. benthamiana* leaves inoculated with *B. cinerea* (Figure 5B and Appendix A). These differences in *BcSAS1* transcript levels between test and control applications were significant at α = 0.05, whereas *BcSAS1* expression levels were the same between control applications of Treatment A and Treatment B (*p* = 0.820), indicating that the reduction in its transcript levels in the test applications was caused by the dsRNAs produced by the HT115/*BcSAS1_Ind_* strain. Moreover, silencing of *BcSAS1* was phenotypically expressed as a parallel reduction in the size of the infection lesions caused by *B. cinerea* on leaves of *N. benthamiana* (Figure 5A,C). Indeed, for Treatment A, a significant 35% reduction in lesion size between test and control application was observed, whereas in Treatment B, this significant reduction was by 25% (Figure 5C and Appendix A). Collectively, these results demonstrate that applications of bacteria that produce dsRNAs against genes in target pathogens can be potentially used for disease control.

## 4. Discussion

Current practices to limit fungal diseases on plants rely heavily on synthetic chemical compounds but their constant use poses a threat to the environment and to human health, and further increases the risk of pathogen resistance development [44]. Host resistance can provide effective control of pathogens but may be limited in some plant species, and pathogens can often overcome resistance [45]. Therefore, alternative methods for disease control are urgently needed to meet the increasing demands in crop yields and quality of produce [44,46,47].

In this study, we exploited the systemic properties of dsRNAs and successfully utilized bacteria as an alternative to HIGS and SIGS methods to deliver RNAi to filamentous fungi. Bacteria have the advantage that they can be cultured in large volumes relatively inexpensively and protect the produced dsRNAs within their cells until their application [48,49]. Our results showed that even after bacterial lysis, the dsRNAs remained resistant to degradation for at least up to 24 h, and partially until 72 h, providing enough time for the dsRNAs to be absorbed by the fungal cells and induce silencing in target genes. The use of bacteria as vectors for delivering RNAi also circumvents the hurdle and limitations of producing and deploying transgenic plants in the field, whereas bacteria can be further engineered to be non-pathogenic microorganisms or to autolyze under selected induced conditions, thus increasing the safety of application [27]. Indeed, the utility of bacteria as delivery vehicles of RNAi has already been demonstrated against insects, nematodes, and even mammalian cell cultures [27,29,31] but, surprisingly, it had not been tested against fungi. We have now bridged this gap and show that bacteria that have been genetically engineered to produce dsRNAs against fungal genes can be used both in living form and as whole-cell autolysates to induce RNAi in fungi. Indeed, our data showed that even a single application of the recombinant bacteria was sufficient to achieve a significant reduction in production of aflatoxins by *A. flavus* or growth of *B. cinerea*, both under in vitro and in vivo conditions. This finding generates new opportunities for developing RNAi-based control methods against fungi of agronomic or medical importance by using non-pathogenic or attenuated bacteria as a dsRNAs delivery system.

In our studies, we have exploited the widely used HT115(DE3) *E. coli* strain as the host bacterium for the production of dsRNAs [29], but other bacteria species could be potentially engineered to deliver RNAi to fungi, including for example bacteria used in biological control, plant endophytic or symbiotic bacteria, and several others [48]. Indeed, symbiotic gut bacteria of insects were recently successfully engineered to deliver RNAi to their insect hosts [50], indicating that the approach is amendable to different bacterial species that can be used to meet the needs of specific applications. To further increase the versatility of the system and make it suitable for diverse applications, we utilized two recombinant expression vectors that enabled the production of dsRNAs in HT115(DE3) under constitutive or IPTG-induced conditions. Specifically, the first vector employed the constitutive *Kan* promoter and allowed the uni-directional transcription of the two complementary RNA strands from the target DNA template, which then anneal post-transcriptionally in the cytoplasm to form dsRNAs. The second vector was based on the bi-directional transcription of the target DNA sequence in a single reaction using the T7 promoter, which is activated upon binding of the T7 RNA polymerase to it. T7 RNA polymerase translation is in-terms dependent on initial IPTG-induction of transcription of the *T7* gene in HT115(DE3) that encodes it. Inducible systems would generally provide higher efficiency, flexibility, and safety in use, as they enable the production and release of dsRNAs from bacteria under user-selected conditions. Although here we used the IPTG-activated T7 RNA polymerase-based expression system, a range of inducible promoters that can be triggered under specific conditions could be employed [51,52]. Constitutive promoters, on the other hand, increase the ease of dsRNA production and could be used when the persistent presence of recombinant bacteria producing dsRNAs is required to achieve a desirable and sustained silencing effect. Although the kinetics of dsRNAs production by the two expression systems was not part of our study, both expression systems were shown to yield sufficient amounts of biologically active dsRNAs that induced silencing in the target fungal genes. However, direct comparisons between the two systems could not be made, as different target sequences were cloned in the recombinant expression vectors, whereas dsRNA production in the IPTG-activated system would vary depending on the time of IPTG addition to the medium and its concentration [53]. Finally, although frequently utilized in RNAi studies, the production of hairpin-structured RNA (hpRNA) was not included in these studies because it was demonstrated that HT115(DE3) is able to process hpRNA into short fragments, in spite of lacking RNaseIII activity [54].

In order to combine the benefits or trans-kingdom transfer and environmental uptake of dsRNAs, we further engineered the HT115(DE3) *E. coli* strain to undergo programmed cellular autolysis induced by endolysin R, following a freeze-thaw cycle [35]. Both living bacteria and their autolysates induced significant silencing in the target fungal genes in a bacteria concentration-dependent manner, suggesting that further increasing bacterial densities could lead to an even higher effect. However, caution should be taken as increasing siRNAs concentrations past an optimum point does not necessarily lead to increased silencing levels and may even have an opposite effect due to saturation of the RNAi machinery [55,56]. Moreover, the silencing levels achieved by the two delivery methods were on par with those obtained with external applications of purified dsRNAs, indicating that even simple applications with living bacteria or their autolysates that bypass the purification of dsRNAs and their biological stabilization by binding to various materials [3,28], can be effective. However, whether one application was more effective than the other (i.e., living bacteria or autolysates) could not be established, since a combination of parameters, including the fungal species and bacterial concentration, had an effect on the observed silencing levels. For instance, while applications under in vitro conditions with living bacteria were seemingly more effective in inducing *AflC* silencing in *A. flavus*, applications with the bacterial autolysates were slightly more effective under in vivo conditions. When examining silencing of *BcSAS1* in *B. cinerea*, then almost the opposite trend was observed, as applications under in vitro conditions of whole-cell autolysates were more effective at some bacterial concentrations, whereas living bacteria seem to have had a bigger effect under in vivo conditions. Such variation in the effectiveness of the two application methods depending on the fungal species could perhaps relate to the way and rate by which the bacterially-produced dsRNAs are taken-up by the fungal cells. Although it is still unclear how exogenous dsRNAs are internalized within fungal cells, several possible mechanisms may be in effect, including passive diffusion through the fungal cell wall, internalization through plasma membrane-associated receptors or protein channels and pores [57], or through the endocytosis or fusion of bacterial extracellular vesicles carrying the dsRNA [58]. It is plausible that the uptake of dsRNAs from the bacterial lysates is mediated by passive or active diffusion through the cell wall and membranes, whereas living bacteria translocate dsRNAs to the fungal cells through the secretion of extracellular vesicles in a way similar to plants [59]. Indeed, several studies have demonstrated that Gram-negative bacteria, including *E. coli*, release extracellular vesicles that contain diverse RNA molecules, including small RNAs (sRNAs) that can modulate gene expression in recipient eukaryotic host cells [24,58,60,61,62]. Irrespectively of the mechanism(s) that may be in effect, differences in their efficacy and utilization among different fungal species could perhaps explain the lack of correlation seen between applications of living or lysed bacteria, and silencing levels achieved in *A. flavus* and *B. cinerea* under in vitro and in vivo conditions.

Finally, an increase in contact time between the fungi and the living bacteria or their autolysates did not produce a clear benefit in terms of enhancing or maintaining silencing levels in *AflC* or *BcSAS1*. However, one might expect such an effect, at least in applications of living bacteria that continuously produce dsRNAs compared with the one-time dose of dsRNAs received by the fungal cells from the bacterial autolysates. A combination of various factors inherent to both the targeted genes and the fungal species could account for this lack of a stable silencing effect over time. Among others, these may include dsRNAs absorption rates, transcript abundance and mRNA turnover rate of the target genes, and the presence of regulatory feedback mechanisms that could restore gene expression upon silencing [63,64]. Indeed, upregulation of gene expression following RNAi is often reported as a cellular response, mainly as a result of saturation of the RNAi machinery [65], which in our case could be caused by the constant supply of dsRNAs. Nonetheless, although the increase in contact time between fungi and the bacteria did not lead to a significant further increase in *AflC* or *BcSAS1* silencing levels, in contrast, it had a positive effect on reducing production of aflatoxins in *A. flavus* and mycelial growth of *B. cinerea* over time. This could possibly be due to the time needed for the inhibition of gene expression to materialize into an observable phenotypic effect, or because silencing reduces protein production at higher levels than mRNA levels, thus having a higher effect on phenotype. Indeed, discordances between mRNA downregulation and reduction in protein production have been documented before, thus leading to analogous disparities between changes in gene expression and phenotype severity [66,67].

Overall, in this study, we demonstrate that genetically engineered bacteria that produce dsRNAs against selected genes in fungi can be effectively used as vectors for delivering RNAi to fungi with high efficiency, thus opening new avenues for disease control.

## 5. Conclusions

The *E. coli* HT115(DE3) strain can produce biologically active dsRNAs against target genes in fungi.The biologically active bacterial dsRNAs were successfully delivered to fungi through two different approaches: (i) treatment with living bacteria and (ii) treatment with bacterial lysates following a programmable cell autolysis step.In vitro applications of dsRNA-producing bacteria show that the degree of the RNAi effect on fungi is positively correlated with the concentration of the bacteria, while the phenotypic changes (i.e., reduction in production of aflatoxins in *A. flavus* and reduction of mycelial growth in *B. cinerea*) are positively correlated with the contact time.In vivo applications of dsRNA-producing bacteria result in a reduction of aflatoxins production by *A. flavus* on corn seeds and a reduction of disease symptoms caused by *B. cinerea* on *N. benthamiana* leaves.

## Figures and Tables

**Figure 1 jof-07-00125-f001:**
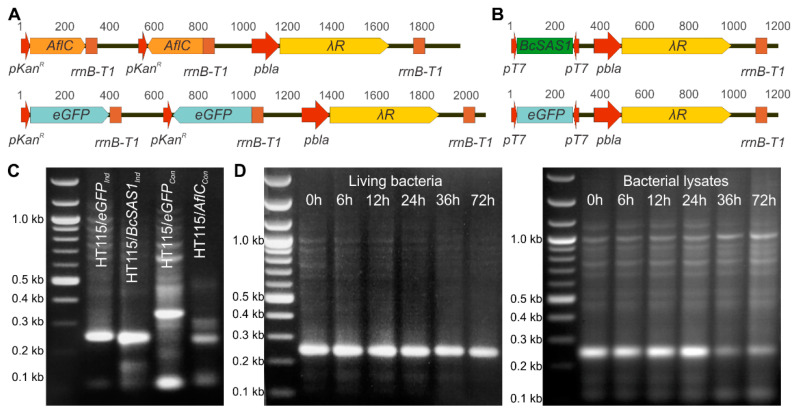
Bacteria can be genetically engineered to produce dsRNAs against target genes in fungi. (**A**,**B**) Schematic representation of the two transcriptional configurations used for the synthesis of dsRNAs against *AflC* in *A. flavus* and *BcSAS1* in *B. cinerea*, respectively as well as for the synthesis of dsRNA against the enhanced green fluorescent protein (eGFP). The full plasmid maps are shown in Appendix A. *pKan^R^*: kanamycin resistance gene; *rrnB-T1:* transcriptional terminator T1 of the *rrnB* gene; *pT7*: T7 RNA polymerase promoter; *pbla*: promoter of the β-lactamase gene; *λR:* phage lambda endolysin gene; *AflC:* Type I polyketide synthase (pksA) in *A. flavus*; *BcSAS1*: Rab family member of small GTPases; *eGFP:* enhanced green fluorescent protein. (**C**) Agarose gel electrophoresis of dsRNAs extracted from the autolyzed HT115/*AflC_Con_*, HT115/*BcSAS1_Ind_*, HT115/*eGFP_Con_*, and HT115/*eGFP_Ind_* strains. Production of dsRNAs in the HT115/*BcSAS1_Ind_* and HT115/*eGFP_Ind_* strains requires the addition of IPTG in the culture medium of the bacteria, which was added at a final concentration of 1.0 mM when cultures reached an OD_600_ ≈ 0.8. (**D**) Stability over time of the dsRNAs produced by the HT115/*BcSAS1_Ind_* strain over time. Stability was examined both in living bacteria cells (left-hand side image) and their whole-cell autolysates (right-hand side image). An equal volume of 5 μL was loaded in the wells of the agarose gel.

**Figure 2 jof-07-00125-f002:**
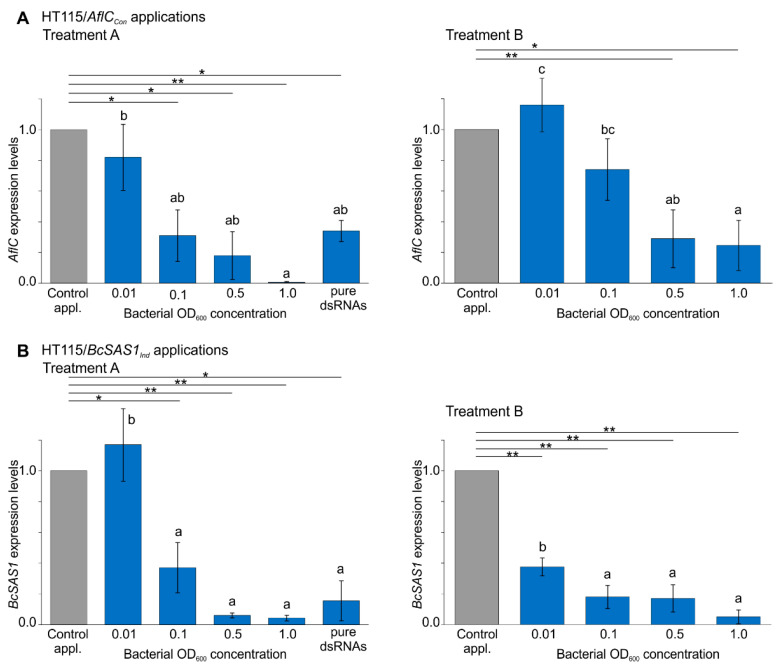
Bacterially-produced dsRNAs induce RNAi in fungi in a bacteria concentration-dependent manner. (**A**,**B**) Expression levels of *AflC* from *A. flavus* (Panel A) and *BcSAS1* from *B. cinerea* (Panel B), in in vitro cultures of the fungi with living (Treatment A) or lysed (Treatment B) *E. coli* HT115(DE3) bacteria producing dsRNAs against *AflC* (HT115/*AflC_Con_*) or *BcSAS1* (HT115/*BcSAS1_Ind_*), respectively (test applications; blue bars). Expression levels are shown relative to the expression of *AflC* and *BcSAS1* when *A. flavus* and *B. cinerea* are co-incubated with the control HT115(DE3) bacteria producing dsRNAs against *eGFP* (HT115/*eGFP_Con_* and HT115/*eGFP_Ind,_* respectively) (control applications; grey bars set to 1.0). Living bacteria or their whole-cell autolysates, obtained from the OD_600_ 0.01, 0.1, 0.5, and 1.0 cultures were added to the fungal cultures, and the expression of *AflC* or *BcSAS*1 was measured at 12 h of co-incubation. For comparison, *AflC* and *BcSAS1* silencing levels achieved by the addition in the in vitro cultures of *A. flavus* or *B. cinerea* of 15 μg (12.5 nM) of purified dsRNAs against each gene, respectively were also evaluated. Significant differences between control and test applications were determined by a one-way ANOVA followed by a Tukey’s HSD test, and levels of significance are indicated above the bars by underlined star symbols (i.e., * *p* < 0.05, ** *p* < 0.01). Differences in the gene silencing efficacy of *AflC* and *BcSAS1* among the test applications were tested by ANOVA and Tukey’s HSD test, and significantly different groups (*p* < 0.05) are indicated by letters above the bars. Error bars in the figure indicate standard deviation (SD) obtained from three biological replicates.

**Figure 3 jof-07-00125-f003:**
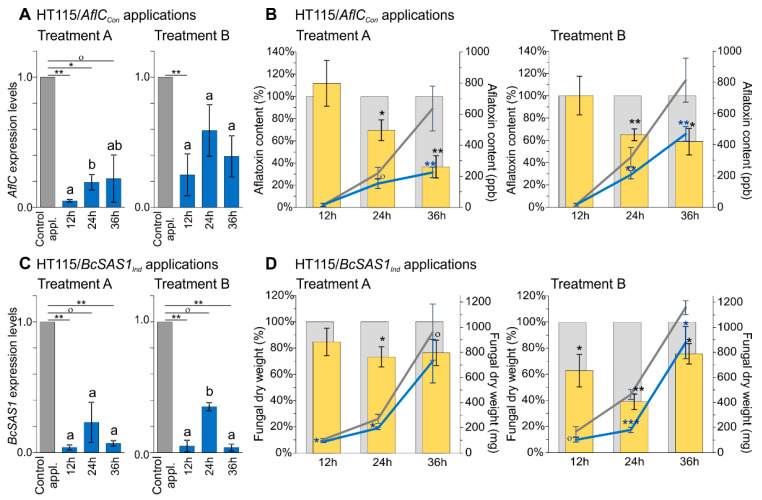
Increased contact times between fungi and dsRNA-producing bacteria have a positive effect on reducing production of aflatoxins in *A. flavus* and mycelial growth in *B. cinerea*. (**A**,**B**) *AflC* expression levels over time (Panel A) and the amount of aflatoxins produced by *A. flavus* (Panel B) in in vitro cultures of the fungus with living (Treatment A) or lysed (Treatment B) *E. coli* HT115(DE3) bacteria producing dsRNAs against *AflC* (HT115/*AflC_Con_*; test application) or *eGFP* (HT115/*eGFP_Con_*; control application). Living bacteria or their whole-cell autolysates, obtained from the OD_600_ 1.0 culture, were mixed with the fungal cultures, and *AflC* expression or the amount of aflatoxins in the co-cultures were measured at 12 h, 24 h, and 36 h. In Panel A, the expression levels of *AflC* in the test applications (blue bars) are shown relative to the control ones (grey bars, set to 1.0). Significant differences between control and test applications were determined by a one-way ANOVA followed by a Tukey’s HSD test, and levels of significance are indicated above the bars by underlined star symbols (i.e., ^o^
*p* < 0.1, * *p* < 0.05, ** *p* < 0.01). Differences in the gene silencing efficacy of *AflC* among the test applications were tested by ANOVA and a Tukey’s HSD test, and significantly different groups (*p* < 0.05) are indicated by letters above the bars. In Panel B, yellow bars indicate the mean relative amount of aflatoxins produced by the fungus in the test relative to control applications, with the latter set to 100% and shown as the background grey bars. The slope analysis in Panel B shows the actual amount of aflatoxins in ppb produced over time by the fungus in control (grey lines) and test applications (blue lines). Error bars in Panels A and B indicate standard deviation (SD) obtained from three biological replicates. Significant differences in the relative mean amount of aflatoxins between control and test applications were determined by one-sample *t*-test, while significant differences in the absolute amount of aflatoxins were determined by a one-way ANOVA followed by a Tukey’s HSD test. Levels of significance are indicated above the bars by blue star symbols (i.e., ^o^
*p* < 0.1, * *p* < 0.05, ** *p* < 0.01). (**C**,**D**) Similar to Panels A and B, but showing *BcSAS1* expression levels (Panel C) and the amount of biomass produced by *B. cinerea* (Panel D) over time in in vitro cultures of the fungus with living (Treatment A) or lysed (Treatment B) *E. coli* HT115(DE3) bacteria producing dsRNAs against *BcSAS1* (HT115/*BcSAS1_Ind_*; test application) or *eGFP* (HT115/*eGFP_Ind_*; control application).

**Figure 4 jof-07-00125-f004:**
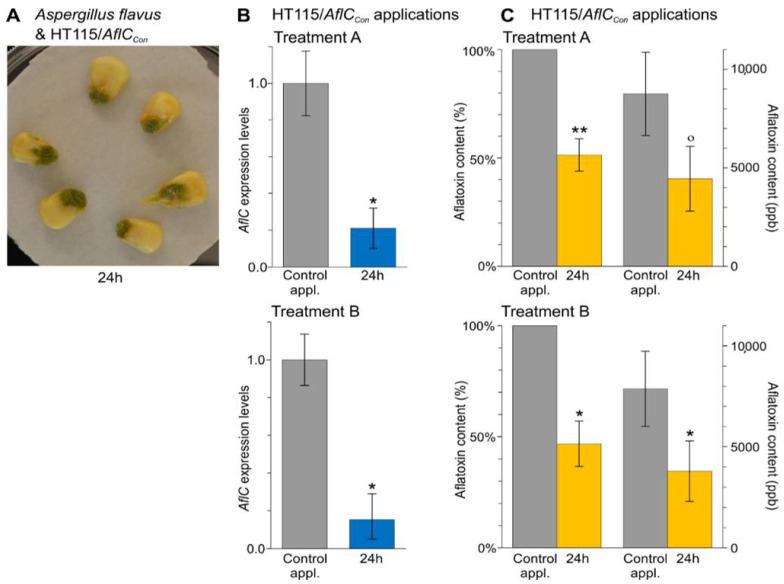
The HT115/*AflC_Con_* strain induces RNAi in *A. flavus* and a reduction in production of aflatoxins under in vivo conditions. (**A**) *A. flavus* inoculations on maize kernels. Spores of the fungus were spot inoculated on maize kernels and 48 h later, 10 μL of the living bacteria or of their whole-cell autolysates, obtained from the OD_600_ 1.0 culture, were applied to the same inoculation spot. All measurements were taken 24 h later. (**B**,**C**) *AflC* expression levels (Panel B) and the amount of aflatoxins produced by *A. flavus* in vivo on maize kernels (Panel C), following its exposure to living (Treatment A) or lysed (Treatment B) *E. coli* HT115(DE3) bacteria producing dsRNAs against *AflC* (HT115/*AflC_Con_*; test application) or *eGFP* (HT115/*eGFP_Con_*; control application). In Panel B, the expression level of *AflC* in the test applications (blue bars) is shown relative to the control applications (grey bars, set to 1.0). In Panel C, charts on the left-hand side show the mean relative aflatoxins content in the maize kernels in the test (yellow bars) relative to the control applications (grey bars, set to 100%) and, on the right-hand side, the actual amount of aflatoxins (in ppb) in the test and the control applications. Error bars in Panels B and C indicate standard deviations (SD) obtained from at least three biological replicates. Significant differences in gene expression and the absolute amount of aflatoxins between control and test applications were determined by a one-way ANOVA followed by a Tukey’s HSD test. Significant differences in the mean relative aflatoxins content were determined by one-sample *t*-test. Levels of significance are indicated above the bars by underlined star symbols (i.e., ^o^
*p* < 0.1, * *p* < 0.05, ** *p* < 0.01).

**Figure 5 jof-07-00125-f005:**
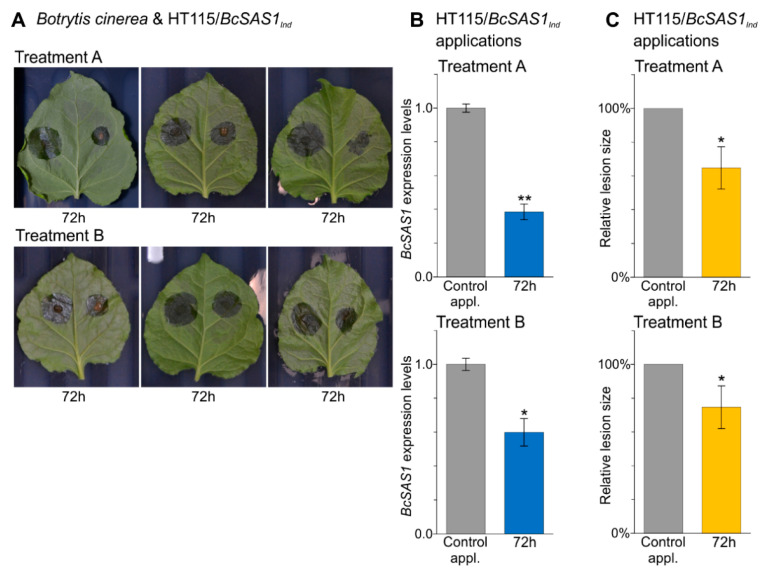
The HT115/*BcSAS1_Ind_* strain induces RNAi in *B. cinerea* and a reduction in fungal growth under in vivo conditions. (**A**) *B. cinerea* inoculations on *N. benthamiana* leaves. Spores of the fungus were spot inoculated on leaves of *N. benthamiana* and 6 h later, 10 μL of the living bacteria or the whole-cell autolysates, obtained from the OD_600_ 1.0 culture, were applied to the same inoculation spots. All measurements were taken 72 h later. (**B**,**C**) *BcSAS1* expression levels (Panel B) and the relative infection lesion size produced by *B. cinerea* in vivo on leaves of *N. benthamiana* (Panel C), following its exposure to living (Treatment A) or lysed (Treatment B) *E. coli* HT115(DE3) bacteria producing dsRNAs against *BcSAS1* (HT115/*BcSAS1_Ind_*; test application) or *eGFP* (HT115/*eGFP_Ind_*; control application). In Panels B and C, *BcSAS1* expression levels and the mean relative lesion size of the infection areas in the test applications (blue bars and yellow bars, respectively) are relative to the control ones (grey bars, set to 1.0 and 100%, respectively). Error bars indicate standard deviations (SD) obtained from at least three biological replicates. Significant differences in gene expression between control and test applications were determined by a one-way ANOVA followed by Tukey’s HSD test, whereas significant differences in the mean relative lesion size were determined by one-sample *t*-test. Levels of significance are indicated above the bars by underlined star symbols (i.e., * *p* < 0.05, ** *p* < 0.01).

## Data Availability

The data that support this study are available from the corresponding author upon reasonable request.

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
