# Peer review of "Targeted Delivery of Gene Silencing in Fungi Using Genetically Engineered Bacteria"

_jof, 2021, doi:10.3390/jof7020125_

Round 1
Reviewer 1 Report
Niño-Sánchez, J. et al. have investigated an RNAi-based approach for fungicidal application using duplex RNAs produced by engineered that target the genes of fungi. This is a quite interesting study and timely given that its direct application in food and environment and human health. The authors thoroughly investigated and the results corroborate the conclusions of the study. A critical reading is warranted, as some typos were found (Freeze-though, amendable, necessities….to name a few) across the manuscript, prior to its publication.
Author Response
We are thankful to the reviewer for her/his time to carefully review our manuscript and we appreciate the positive feedback. We also thank the reviewer for spotting typos in our manuscript and bringing those to our attention. We apologize for these typos, which we have now corrected after carefully proofreading our manuscript. Some additional edits were also made to the text, including adding some more background information in the introduction and citing more articles.
Once more we kindly thank the reviewer for evaluating our manuscript.
Sincerely,
Ioannis Stergiopoulos
Reviewer 2 Report
Dear Editor,
Thanks for sending the manuscript for reviewing. The current work focuses on gene silencing in fungi.
The molecular work sees to be interesting however, the authors check only for the total aflatoxins. I would ask about cyclopiazonic acid. Have the authors checked that?
For ELISA test, the calibration curve was from 4 until 40 ppb (line 273) however in figures 3 and 4 I can see that AFs concertation is up to 1000 ppb. How did you measure this concentration which is out of your calibration curve ??
Author Response
Dear Editor,
Thanks for sending the manuscript for reviewing. The current work focuses on gene silencing in fungi.
Response: We thank the reviewer for evaluating our manuscript and for raising some interesting questions. Following, we provide answers to these questions, which we hope that the reviewer will find satisfactory. On the evaluation form of the manuscript, the reviewer has also pointed out that all sections of the manuscript must be improved. However, it is unclear to us why this should be the case since no arguments, examples, or a justification was provided in support of the reviewer’s evaluation. Our manuscript contains very detailed and precise descriptions of methods and results, with all the data provided in tables and figures so that readers can reproduce our methods and results if needed. Our rationale for each experiment is clearly stated at the start of each experimental section, the study design exemplified and the conclusions are extracted straight from the data. For these reasons, we respectfully disagree with the sweeping conclusion that the entire manuscript, needs to be improved.
The molecular work sees to be interesting however, the authors check only for the total aflatoxins. I would ask about cyclopiazonic acid. Have the authors checked that?
Response: Thank you for your comment. We have not tested whether production of cyclopiazonic acid (CPA) is affected by the application of dsRNAs against AflC in A. flavus. This is because CPA production should not be affected by our experiments. Specifically, CPA is biosynthesized from a pathway that is adjacent but distinct to that of aflatoxins (Chang et al. 2009). Therefore, inhibiting the biosynthesis of aflatoxins through silencing of AflC, which is not a regulatory gene, will not affect CPA production. Also, based on BLAST results, the 250 bp fragment that was used for silencing of AflC does not align with the sequence of cpaA, the PKS–NRPS gene involved in the biosynthesis of the backbone structure of CPA (Tokuoka et al. 2008, Seshime et al. 2009), or with other sequences within the CPA gene cluster (JN712209.1). This means that our 250bp fragment will not induce silencing of the genes in the CPA cluster. Finally, AflC silencing did not result in a reduction of fungal growth, and thus CPA production should not be affected as a result of reduced growth.
References:
Chang PK, Ehrlich KC, Fujii I. Cyclopiazonic acid biosynthesis of Aspergillus flavus and Aspergillus oryzae. Toxins. 2009 Dec;1(2):74-99.
Seshime Y, Juvvadi PR, Tokuoka M, Koyama Y, Kitamoto K, Ebizuka Y, Fujii I. Functional expression of the Aspergillus flavus PKS–NRPS hybrid CpaA involved in the biosynthesis of cyclopiazonic acid. Bioorganic & medicinal chemistry letters. 2009 Jun 15;19(12):3288-92.
Tokuoka M, Seshime Y, Fujii I, Kitamoto K, Takahashi T, Koyama Y. Identification of a novel polyketide synthase–nonribosomal peptide synthetase (PKS–NRPS) gene required for the biosynthesis of cyclopiazonic acid in Aspergillus oryzae. Fungal Genetics and Biology. 2008 Dec 1;45(12):1608-15.
For ELISA test, the calibration curve was from 4 until 40 ppb (line 273) however in figures 3 and 4 I can see that AFs concertation is up to 1000 ppb. How did you measure this concentration which is out of your calibration curve ??
Response: Thank you for raising this point. In the revised manuscript, we added more details in the Materials and Methods section that refers to the quantification of aflatoxins in the samples (i.e. section 2.6 ‘Analysis of total aflatoxin by ELISA”) that we hope will clarify why the measured concentration of aflatoxin appeared to be outside of the calibration curve of 4 to 40 ppb of the ELISA kit. Briefly, prior to measuring aflatoxin concentration using the ELISA AgraQuant® Total Aflatoxin Test Kit, we made several serial dilutions of the samples (i.e., 1:0, 1:10, 1:100 and 1:1000) in order to ensure that one of the dilutions would fall within the aflatoxin concentration range covered by the standards (i.e., 0, 4, 10, 20, 30 and 40 ppb) provided with the manufacture’s kit. These dilutions were then run with the standards in an ELISA test, and for each sample we chose the dilution whose absorbance at OD450 was in the range of the calibration curve. Then, we carried out the ELISA quantification step with the selection of diluted samples that could be satisfactory interpolated in the calibration curve. To calculate the final aflatoxin concentration in the original samples, we took the dilution factor into account, and thus although our final readout is above the measurable range, the experiment was conducted in the correct range.
Reviewer 3 Report
Ingestion of dsRNA expressed within E. coli bacteria has been a useful method to induce RNA interference in model systems and first shown in the nematode, C. elegans. In this work, the authors demonstrate RNAi using the same bacteria, E. coli HT115(DE3), as used for C. elegans feeding-based RNAi, to induce silencing of genes in two fungal species, A. flavus and B. cinerea.
The authors tested various configurations for production of dsRNA, including separate sense and antisense expressing transgenes, and a single transgene flanked by the T7 RNA polymerase sequence in inverted orientation.
The authors further engineered HT115 to express AflC in A. flavus and BcSAS1 in B. cinerea. They further enhanced the HT115 dsRNA release with an inducible autolysis system, induction of lambda endolysin under the control of the beta lactamase promoter.
Exposure of fungi to the bacteria triggered gene silencing, dependent on bacterial concentration, and at highest concentrations, better than purified dsRNA at high concentration. Interestingly, the effect is strongest with 12h of exposure, and longer treatments do not result in as significant a reduction in target mRNA levels compared to controls. The authors also measured phenotypic effects from the knockdown of the genes in each species, i.e. aflatoxin production and mycelial growth. Here the effects were stronger over time, consistent with a time lag of mRNA knockdown and hence protein knockdown.
The authors next test co-inoculation of fungi and bacteria onto maize seeds and tobacco leaves. They were able to measure significant reductions of targeted transcripts. In the case of A. flavus, aflatoxin content was reduced by about 50%, and for B. cinerea, size of lesions was reduced by about 40%-50%.
The paper is written well and includes highly detailed descriptions of the methodology and a highly detailed set of supplementary data. The observance of both mRNA reduction and expected phenotypes relative to controls support the occurrence of RNAi mediated by the treatments with engineered E. coli bacteria. The weakness of the effects, compared to the robust RNAi response seen in, say, C. elegans, is perhaps a bit discouraging, but this should not be a reason to avoid these methods in establishing genetic cause-effect relationships in other contexts. For example, it could be a way of screening candidate genes in a low-cost, high-throughput way before making chromosomal mutants by CRISPR/Cas9. Overall, then, the experiments are well-designed and should enable others to replicate the same RNAi effects in their systems, and the work would be useful for that purpose. The paper can be accepted with minor corrections as noted below.
line 127 – concertation – concentration
line 203 – insert citation of the 2-delta delta Ct method, i.e. Livak & Schmittgen (2001), https://doi.org/10.1006/meth.2001.1262
line 321 – in an invert orientation – in an inverted orientation
lines 145, 696, 731 – HT155 – should be HT115
Author Response
We thank the reviewer for the time and effort that she/he has invested in reviewing this manuscript and for acknowledging the research presenting in it. We also thank the reviewer for bringing some typos to our attention. All authors have carefully proofread the manuscript and corrected these and other typos that came into our attention. We have also modified the introduction of the manuscript by adding some more background information and citing more articles.
We thank again the reviewer for evaluating our manuscript and for providing feedback.
Sincerely,
Ioannis Stergiopoulos
Response to comments and suggestions:
line 127 – concertation – concentration
Response: Thank you for bringing this spelling mistake to our attention. It has now been corrected.
line 203 – insert citation of the 2-delta delta Ct method, i.e. Livak & Schmittgen (2001), https://doi.org/10.1006/meth.2001.1262
Response: Thank you for bringing this work to our attention. The reference has been added to the manuscript.
line 321 – in an invert orientation – in an inverted orientation
Response: Thank you for bringing this to our attention. The grammatical error has been corrected.
lines 145, 696, 731 – HT155 – should be HT115
Response: Thank you for bringing this typo to our attention. The typo has been corrected through the text.
Round 2
Reviewer 2 Report
The authors could address all the rasied commetns, however there are some mistakes.
line 88 , it is aflatoxins NOT aflatoxin unless you states and certain type>> please correct.
please add more info in aflatoxins, this is the most carcinogenic natural substance in the world!!
I suggest add. Contamination of crops, animal feed and human food with aflatoxins has been very documented (references).
Adballah et la., 2017 Occurrence of multiple mycotoxins and other fungal metabolites in animal feed and maize samples from Egypt using LC‐MS/MS
Author Response
The authors could address all the rasied commetns, however there are some mistakes.
Response: We thank the reviewer for re-evaluating our manuscript and being supportive of the modifications made in its previous version. We hope that we have satisfactory addressed the reviewer’s comments in this version of the manuscript as well.
line 88 , it is aflatoxins NOT aflatoxin unless you states and certain type>> please correct.
Response: Thank you for pointing this out. Indeed, there are several types of aflatoxins and since we have not measured a specific one (e.g. AFB1, or AFB2, etc), references to ‘aflatoxin’ were changed as needed in the text to ‘aflatoxins’ (i.e. plural).
please add more info in aflatoxins, this is the most carcinogenic natural substance in the world!!
Response: Thank you for the suggestion. The reviewer is correct. Aflatoxins are from the most potent carcinogens and to highlight this, we have added a paragraph in the introduction (lines 54-67).
I suggest add. Contamination of crops, animal feed and human food with aflatoxins has been very documented (references).
Adballah et la., 2017 Occurrence of multiple mycotoxins and other fungal metabolites in animal feed and maize samples from Egypt using LC‐MS/MS
Response: Thank you for the suggestion. The following three review papers were cited in the text in reference to aflatoxins, as they are more pertinent to work described in the manuscript. We hope that the reviewer finds these references satisfactory.
Kumar, P.; Mahato, D.K.; Kamle, M.; Mohanta, T.K.; Kang, S.G. Aflatoxins: a global concern for food safety, human health and their management. Front. Microbiol. 2017, 7, 2170.
Mahato, D.K.; Lee, K.E.; Kamle, M.; Devi, S.; Dewangan, K.N.; Kumar, P.; Kang, S.G. Aflatoxins in Food and Feed: An Overview on Prevalence, Detection and Control Strategies. Front. Microbiol. 2019, 10.
Robens, J.F.; Richard, J.L. Aflatoxins in animal and human health. Rev. Environ. Contam. T 1992, 127, 69-94.